

# A new method of gall mite management: application of artificial defoliation to control *Aceria pallida*

Jianling Li, Sai Liu, Kun Guo, Haili Qiao, Rong Xu, Changqing Xu and Jun Chen

Institute of Medicinal Plant Development, Chinese Academy of Medical Sciences and Peking Union Medical College, Beijing, China

## ABSTRACT

Artificial defoliant is widely applied to cotton to facilitate mechanical harvesting and successfully controls leaf diseases by blocking pathogen epidemical cycles; however, this technique is rarely used to control herbivores. Because many eriophyoid mites live and reproduce in galls, the control of these mites by pesticides is usually limited. However, the abscission of galled foliage is lethal to tiny mites with low mobility. Therefore, artificial defoliation should be effective in controlling gall mites. Here, the effects of defoliant on the control of the goji berry *Lycium barbarum* L. gall mite *Aceria pallida* Keifer were compared with those of pesticides under field conditions over 3 years. Our results showed that artificial defoliation enabled almost complete defoliation and timely refoliation. *A. pallida* galls fell off with the defoliation, and then regenerated foliage escaped from mite attack. After defoliant application, the densities of mite galls decreased by 84.1%, 80.3% and 80.3% compared with those found in the pesticide (undefoliated) treatment in 2012, 2013 and 2014, respectively. Artificial defoliation achieved much better control of gall mites than pesticides.

## INTRODUCTION

Phytophagous mites cause serious direct damage to economically important plants by sucking plant sap (*Van Leeuwen et al., 2010*; *Marcic, 2012*) and lead to indirect damage as vectors of plant pathogens (*Andret-Link & Fuchs, 2005*; *De Lillo et al., 2018*). Chemical control is usually efficient in suppressing the damage caused by free-living mites, which live on the surface of plant tissues (*Marcic, 2012*; *Van Leeuwen et al., 2014*). However, some species, especially eriophyoid mites, induce galls on plant tissues as refuges in which these mites spend most of their life cycle; thus, pesticide control of such species is always limited (*Childers, Easterbrook & Solomon, 1996*; *Navia et al., 2010*; *Van Leeuwen et al., 2010*).

For eriophyoid mites with tiny bodies (adult body length averaging approximately 200 μm) (*Lindquist, 1996*), passive long-distance dispersal mainly depends on wind, which is inefficient and poses a high risk for host-specific mites to land on suitable plants (*Lindquist & Oldfield, 1996*; *Michalska et al., 2010*). Active dispersal by slow walking only occurs over relatively short distances, mainly within the same plant or between plants touching each

Corresponding authors
Changqing Xu, cqxu@implad.ac.cn, cqxu@hotmail.com
Jun Chen, jchen@implad.ac.cn

other (*Michalska et al., 2010*). Leaf abscission takes the mites too far to return to the host plant by ambulation (*Sabelis & Bruin, 1996*); thus, the defoliation of gall foliage is fatal to gall mites. In practice, farmers often prune the infested leaves or branches to decrease gall mite damage in addition to applying pesticides (*Oldfield & Proeseler, 1996*; *Duso et al., 2010*). Although pruning galled tissues is always considered effective in controlling gall mites, this method is inefficient and costly. In this study, artificial defoliation is proposed as a feasible and effective method of decreasing the damage caused by gall mites.

Artificial defoliation is widely applied to cotton to facilitate mechanical harvesting and often used to simulate defoliation by herbivory to study plant responses (*Kulman, 1971*; *Lee & Morton, 2003*; *Quentin et al., 2010*). Many studies have been published about the effects of artificial defoliation on plant physiology, yield and quality (*Reichenbacker, Schultz & Hart, 1996*; *Faircloth et al., 2004*; *Eyles et al., 2013*; *Mo et al., 2018*). Additionally, artificial defoliation has been shown to be effective in preventing leaf disease caused by *Colletotrichum gloeosporioides* Penz. and *Oidium heveae* Steinm. in *Hevea* rubber trees by accelerating defoliation and refoliation to disrupt pathogen epidemical cycles (*Rao, 1971*; *Guyot et al., 2001*). Few studies have considered the control effects of artificial defoliation on phytophagous pests, especially gall mites, which are difficult to control with pesticides. Knowledge of how artificial defoliation affects gall mites may provide a new approach for controlling these kinds of pests.

In this study, the eriophyoid mite *Aceria pallida* Keifer (Eriophyoidea) and its host goji berry bush *Lycium barbarum* L. (Solanaceae) were used as a model system. The gall mite is a predominant pest of the goji berry bush, which is among the most widely cultivated medicinal herbs in China (*Xu et al., 2014*). The mite mainly feeds on foliage, leading to tissue deformation and gall formation (Fig. 1), and decreases in plant production. Because gall mites reproduce and live in galls, the period to effectively control these mites is usually confined to the time when they are emigrating from galls to invade other tissues (*Childers, Easterbrook & Solomon, 1996*; *Hrudová & Šafránková, 2017*). Eriophyoid mites reproduce by parthenogenesis, their generations overlap considerably, and hundreds of mites of different stages live in each gall (*Oldfield & Michalska, 1996*; *Michalska et al., 2010*). Pesticides are frequently utilized to suppress mite population growth to ensure the protection of these bushes; however, pesticide abuse not only increases mite resistance but also causes pollution to the goji berry fruit and environment (*Xu et al., 2014*). As pesticide contamination is currently a bottleneck in the export of goji berry in China, safe and effective methods of controlling gall mites are urgently needed in production systems to decrease the use of pesticides (*Xu et al., 2014*; *Yao et al., 2018*).

Similar to many other deciduous trees, goji berry bushes undergo defoliation twice per year. The first defoliation occurs after harvest in July and during the growing season, and the second defoliation occurs in November to allow for overwintering survival (*Li et al., 2018*). Defoliation in July is partial and prolonged and proceeds simultaneously with refoliation. Adults of *A. pallida* have sufficient time to emigrate from galled foliage to regenerated foliage. Consequently, the damage caused by the gall mite reappears in autumn and causes serious damage to production. Moreover, the large overwintering population increases the difficulty of controlling the mite in the next year. In this study, it is proposed

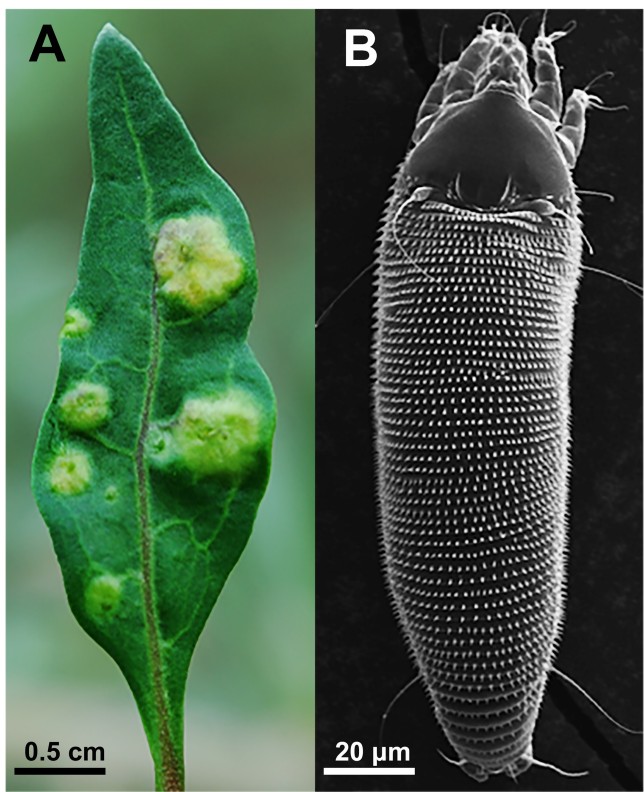

**Figure 1** (A) Galls induced by *A. pallida* on leaf; and (B) adult *A. pallida* observed using a scanning electron microscope.

that the renewal of foliage period in July would be an appropriate time to apply artificial defoliation to control *A. pallida*. Here, the control effect of a defoliant on *A. pallida* was compared with that of pesticides under field conditions after harvest in July throughout a period of 3 years.

## MATERIALS AND METHODS

### Study site

The study was conducted in an experimental site of 2,520 m$^2$ (28 m width, 90 m length), located in Zhongning (37°29′N and 105°42′E), Ningxia Hui Autonomous Region, China, throughout 3 years (2012, 2013 and 2014) from July to November. The site was planted with 840 bushes (14 columns and 60 rows) with a 2 m inter-row spacing and 1.5 m intra-row spacing in 2001. The crown diameter (approximately 1.4 m) and height (approximately 1.5 m) of these bushes were similar.

### Experimental design

The experiments with defoliant and pesticide (undefoliated) treatments were conducted during the self-renewal of foliage period in July. Prior to the study, the experimental site was treated with pesticides according to local pesticide usage. Based on the methods

**Table 1  Information and applied doses of defoliant and pesticides.**

| Product | | Manufacturer | Active ingredient concentration and formulation | Applied doses (mg AI/kg) |
|---|---|---|---|---|
| Defoliant | Dropp ultra® | Bayer Crop Science, Leverkusen, Germany | 540 g/L (360 g/L diuron and 180 g/L thidiazuron) suspension concentrate | 72 |
| Pesticides | Abamectin | North China Pharmaceutical Group Aino Co., Ltd, Shijiazhuang, China | 1.8% emulsifiable concentrate | 15 |
| | Imidacloprid | Bayer Crop Science, Leverkusen, Germany | 200 g/L soluble concentrate | 100 |
| | Chlorpyriphos | Dow AgroSciences, Indianapolis, USA | 40% emulsion in water | 400 |
| | Acetamiprid | Hebei Weiyuan Biological and Chemical Co., Ltd., Shijiazhuang, China | 20% soluble concentrate | 40 |
| | Spinetoram | Dow AgroSciences, Indianapolis, USA | 60 g/L suspension concentrate | 30 |
| | Sulphur | Hebei Shuangji Chemicals Co., Ltd., Xinji, China | 50% suspension concentrate | 1,000 |
| | Azadirachtin | Chengdu Green Gold Hi-Tech Co., Ltd., Chengdu, China | 0.3% emulsifiable concentrate | 6 |
| | Matrine | Jiangsu Fengshan Group Co., Ltd., Yancheng, China | 0.3% soluble concentrate | 6 |

outlined by *Lawal (2014)*, the experimental site was equally divided into ten plots, with each plot consisting of 84 bushes (seven columns and 12 rows); and the two treatments of 5 plots each were arranged in a completely randomized design each year. All bushes of a plot received the same treatment, and the outside rows (34 bushes) were considered buffer areas and were not sampled. One defoliant and eight pesticides were utilized to manage the gall mite, and they were applied by a mechanical sprayer (SP-50, 21–40 kg/cm$^2$, Shanghai Panda Machinery Co., Ltd, China) (Table 1). Defoliant without pesticide was sprayed only one time in each defoliant plot each year. To compare the control effect of the defoliant and pesticides (including chemical, mineral and biological pesticides) on the gall mite, pesticides were sprayed two, four and three times based on the local use of chemical pesticide in 2012, 2013 and 2014, respectively (Table 2).

To study the effects of defoliant and pesticides on defoliation and refoliation, two out of 50 bushes were chosen randomly in different columns and rows in each plot in 2012. Then, four branches per bush at approximately 20 cm long from the tip (approximately 30 leaves per branch before defoliant application) at different orientations were tagged to record the number of old foliage and regenerated foliage at 0, 1, 3, 7 and 13 days after defoliant application. To study the effects of the defoliant on the dynamics of galls, two bushes and four branches per bush were chosen to record the number of *A. pallida* galls twice per month using the abovementioned sampling methodology from 2012 to 2014.

## Statistical analysis

The statistical software SPSS version 21.0 (IBM, Armonk, NY, USA) was used for the statistical analyses. Significant differences in the density of leaves at different orientations were analyzed using a one-way ANOVA followed by Tukey's HSD tests. Significant differences in the density of leaves in different treatments were analyzed using independent

**Table 2  Application information for the defoliant and pesticides from 2012 to 2014.**

| Treatment | Application time | | Ingredient |
|---|---|---|---|
| Defoliant | 2012 | 23-Jul | Dropp ultra® |
| | 2013 | 12-Jul | |
| | 2014 | 20-Jul | |
| Pesticides | 2012 | 2-Aug | Abamectin + chlorpyriphos |
| | | 4-Sep | Abamectin + acetamiprid + imidacloprid |
| | 2013 | 12-Jul | Abamectin + chlorpyriphos |
| | | 24-Jul | Abamectin + chlorpyriphos + imidacloprid |
| | | 26-Aug | Abamectin + chlorpyriphos |
| | | 5-Sep | Abamectin + acetamiprid |
| | 2014 | 20-Jul | Spinetoram + azadirachtin |
| | | 1-Aug | Spinetoram + azadirachtin + sulphur |
| | | 12-Aug | Azadirachtin + matrine + sulphur |

sample $t$-tests. Significant differences in the dynamics of galls were analyzed using a repeated-measures ANOVA.

# RESULTS

## Effects of defoliant and pesticides on the defoliation and refoliation of foliage

Before defoliant application in 2012, the densities of foliage at different orientations were not significantly different ($F_{7,32} = 1.234$, $P = 0.313$) (Fig. S1). After defoliant application, the foliage fell off much more rapidly and completely and more leaves sprouted in time (Table 3). Three days after defoliant application, more than 90% (94.4%) of the old leaves had fallen off in the defoliant plots; 10.5% had fallen off in the pesticide plots ($t_4 = -32.895$, $P < 0.001$); and none of the foliage regenerated in the two treatments. Seven days after defoliant application, almost all the old foliage (97.1%) had dropped and $5.3 \pm 2.71$ new foliage per branch had sprouted out. However, only 25.4% of the old foliage had defoliated, and no foliage germinated in the pesticide plots. On the 13th day after defoliant application, less than half (41.6%) of the old foliage had dropped and $2.1 \pm 1.44$ new foliage per branch emerged in the pesticide plots. Thus, refoliation and defoliation proceeded simultaneously in the pesticide plots. The number of regenerated leaves in the defoliant treatment was up to 35 times greater than that in the pesticide treatment ($t_4 = 15.223$, $P < 0.001$).

## Effects of defoliant and pesticides on the density of galls

In the pesticide treatment, different kind, combination and application date of pesticides did not effectively prevent the gall mite infestation over 3 years (Figs. 2A–2C). The dynamics of galls followed similar patterns. In July, the number of galls fluctuated slowly with the prolonged renewal of foliage. With the refoliation in August, adult mites migrated from old to young foliage and the gall number increased rapidly. The density of galls reached its peak in September (2012: $22.1 \pm 4.66$; 2013: $16.2 \pm 7.31$) (Figs. 2A and 2B) or October (2014: $21.3 \pm 7.37$) (Fig. 2C). After that, mite galls fell off with the defoliation with the onset of

Table 3 Number of (A) old and (B) new foliage per branch in the defoliant treatment and pesticide treatment after defoliant application in 2012.

| Source | Treatment | Days after defoliant application | | | | |
|---|---|---|---|---|---|---|
| | | 0 | 1 | 3 | 7 | 13 |
| Old foliage | Defoliant | $35.5 \pm 3.92$ | $20.2 \pm 4.83$ | $2.0 \pm 1.30$ | $1.0 \pm 0.63$ | 0 |
| | Pesticides | $31.2 \pm 3.73^{ns}$ | $32.1 \pm 3.59^{**}$ | $27.9 \pm 1.18^{***}$ | $23.2 \pm 2.96^{***}$ | $18.2 \pm 1.72^{***}$ |
| New foliage | Defoliant | 0 | 0 | 0 | $5.3 \pm 2.71^{*}$ | $72.9 \pm 10.29^{***}$ |
| | Pesticides | 0 | 0 | 0 | 0 | $2.1 \pm 1.44$ |

Five replications were performed for each treatment, and two bushes were selected in each replication. Error bars are ±SD. *, ** and *** indicate significant differences between the defoliant and pesticide treatments on the same day, i.e., $P < 0.05$, 0.01 and 0.001, respectively. ns indicates no significant difference on the same day, i.e., $P > 0.05$.

winter. Adult mites migrated from galls to their hibernation sites to ensure overwintering survival (*Liu et al., 2016*).

However, in the defoliant treatment, more than 90% of mite galls fell off within 11 days after defoliant application over the course of 2012, 2013 and 2014 (Figs. 2A–2C). Because most galls had defoliated with the abscission of foliage (Table 3), few mites survived and caused serious damage to plants. The fluctuation of galls was stable at a low density (Figs. 2A–2C). Throughout the investigation period, the mean densities of galls in the defoliant plots were decreased by 84.1% (Fig. 2D), 80.3% (Fig. 2E) and 80.3% (Fig. 2F) compared with those in the pesticide plots in 2012 ($F_{1,4} = 43.917$, $P < 0.001$), 2013 ($F_{1,4} = 19.969$, $P = 0.002$) and 2014 ($F_{1,4} = 14.673$, $P = 0.005$) (Table 4), respectively. Artificial defoliation achieved better control effects on the gall mite than pesticides by blocking nutrient supply.

## DISCUSSION

Our results showed that increasing the frequency of chemical pesticides could not effectively prevent gall mite infestation and high doses of natural pesticides did not achieve better control of the mite than chemical pesticides and defoliant, although natural pesticides are often considered to be environmentally friendly and easily degradable (*Copping & Menn, 2000*). The study demonstrated that artificial defoliation, a new management method for controlling gall mites, was much more effective than chemical and natural pesticides in preventing eriophyoid mites. The results showed that artificial defoliation facilitated the abscission of old foliage and stimulated timely refoliation. Galls caused by *A. pallida* fell off with the defoliation of galled foliage. When new foliage emerged, almost all the old foliage with galls had been defoliated, and the residual number of gall mites on bushes was too low to cause serious damage. However, since galls provide shelter to mites and systemic pesticides are lacking, neither chemical pesticides nor natural pesticides could effectively prevent the gall mite infestation.

In contrast with other herbivorous mites that hibernate on host plants (*Krantz & Lindquist, 1979*; *Michalska et al., 2010*; *Walter & Proctor, 2013*), *A. pallida* is a phoront that is obligately phoretic on the psyllid *Bactericera gobica* for survival in the winter (*Liu et al., 2016*; *Li et al., 2018*). Although our results confirmed that artificial defoliation was effective in controlling the gall mite, the effect on the psyllid was unknown. Because the psyllid feeds and breeds exclusively on foliage (*Li et al., 2018*), the defoliation and desiccation of foliage

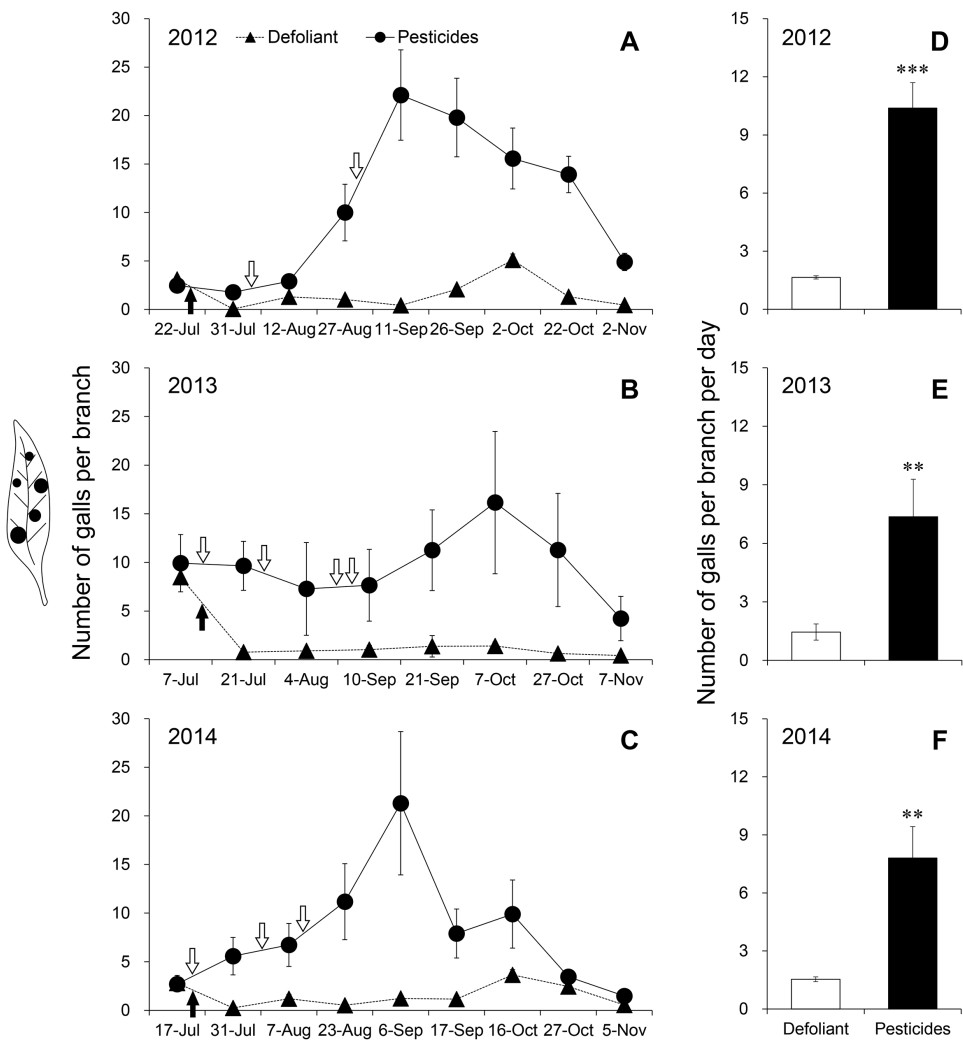

**Figure 2** **Dynamics of galls in the defoliant treatment and pesticide treatment in (A) 2012, (B) 2013 and (C) 2014. Number of galls per branch per day in (D) 2012, (E) 2013 and (F) 2014.** Black arrows indicate the time of defoliant application and white arrows with black outline indicate the time of pesticide application. Five replications were performed for each treatment, and 2 bushes were selected in each replication. Error bars are ±SE. ** and *** indicate significant differences between the defoliant and pesticide treatments, i.e., $P < 0.01$ and $0.001$, respectively.

should be effective in the control of psyllid eggs and their inactive nymphs by blocking the insect's nutrient supply. However, the generations of these pests overlap considerably, and adults with wings may have migrated from the defoliated plots to other areas where food is available. Therefore, the effect of artificial defoliation on the control of pests with high mobility will always be limited. A combination of defoliant and pesticides should be more effective than defoliant alone in controlling these foliage pests and needs to be studied further.

Plant galls are abnormal vegetative growths in plant tissue, and they are most often observed on foliage (approximately 65%) and mainly induced by insects and mites (insects

**Table 4  Results of the analyses of gall dynamics in which comparisons of the number of galls were performed for dates, treatments and their interaction over 3 years of experimentation.** Five replications were performed for each treatment, and 2 bushes were selected in each replication.

| Year | Source | df | Mean square | F | P-value |
|------|--------|-----|-------------|------|---------|
|      | Date | 8 | 586.230 | 11.616 | <0.001 |
| 2012 | Treatment | 1 | 1717.498 | 43.917 | <0.001 |
|      | Date × Treatment | 8 | 508.422 | 10.074 | <0.001 |
|      | Date | 7 | 242.973 | 1.098 | 0.348 |
| 2013 | Treatment | 1 | 1216.956 | 19.969 | 0.002 |
|      | Date × Treatment | 7 | 193.424 | 0.874 | 0.416 |
|      | Date | 8 | 399.217 | 3.876 | 0.048 |
| 2014 | Treatment | 1 | 883.475 | 14.673 | 0.005 |
|      | Date × Treatment | 8 | 421.947 | 4.097 | 0.042 |

+ mites: approximately 70%) (*Mani, 1964*; *Abrahamson & Weis, 1987*). Although gall makers rarely cause destructive damage to host plant growth (*Sabelis & Bruin, 1996*; *Stone & Schönrogge, 2003*), some of them cause serious damage to economic plant production. For example, the gall wasp *Dryocosmus kuriphilus* Yasumatsu could reduce the yield of *Castanea sativa* Mill. by as much as 80% (*Battisti et al., 2014*); the gall mite *Aceria rhodiolae* (Canestrini) could decrease the medicinal quality (salidroside) of *Rhodiola rosea* L. by over 50% (*Beaulieu et al., 2016*). Gall-maker larvae acquire nutrition and shelter from plant galls to complete their development (*Price, Fernandes & Waring, 1987*; *Stone & Schönrogge, 2003*); therefore, the defoliation and desiccation of plant galls is deadly to these arthropod herbivores. Because defoliants (tribufos, thidiazuron, ethephon, etc.) can facilitate timely defoliation, we suggest that defoliant application may be effective in the control of other foliage gall-forming pests and not merely goji berry gall mite, by blocking nutrient supply.

Our results showed that defoliant application enabled not only defoliation of goji berry bushes but also quick refoliation. Previous publications have demonstrated that refoliation as a defoliation-induced response of trees represents compensatory regrowth by depleting stored plant reserves (*Kosola et al., 2001*; *Lasseur et al., 2007*; *Erbilgin et al., 2014*; *Nakajima, 2018*). Severe defoliation commonly has negative effects on the growth and reproduction of trees. *Reichenbacker, Schultz & Hart (1996)* reported that the height, diameter and biomass of *Populus* clones decreased significantly with increasing defoliation. *Jetton & Robison (2014)* documented that severe defoliation caused significant reductions in sweetgum *Liquidambar styraciflua* L. stem growth and biomass accumulation. Similarly, *Milbrath (2008)* found that increasing frequencies of severe defoliation caused greater reductions in biomass and seed production of *Vincetoxicum rossicum* (Kleopow) and *V. nigrum* (L.). These detrimental influences induced by defoliation can be alleviated by supplemental nutrients. For example, N fertilization can reverse the negative influence of defoliation on *Populus × canadensi* cv Eugeneii diameter growth (*Kosola et al., 2001*), and N, P, K fertilizer can alleviate the reductions of *P. tremuloides* (Mich.) biomass and leaf non-structural carbohydrate concentrations under repeated defoliation (*Erbilgin et al., 2014*). However, in some cases, artificial defoliation is favourable to the growth of

trees. According to the report by *Guyot et al. (2001)*, artificial defoliation increased rubber production of *H. brasiliensis* (Willd. ex A. Juss.) by blocking leaf fall disease epidemical cycles. Although our results showed that artificial defoliation was effective in controlling gall mites and more regrowth leaves were quickly generated following defoliation, the plants not only lost their photosynthetic capacity during defoliation but also the resources, most notably nitrogen, contained in the leaves (*Aerts, 1996*; *Eckstein, Karlsson & Weih, 1998*; *Kosola et al., 2001*). Therefore, such a severe method might be detrimental to the growth of goji berry bushes over a period of years. Further research is required to reveal the potential long-term effects of artificial defoliation on the growth and production of goji berry bushes and promote quick restoration.

## CONCLUSIONS

Artificial defoliation as a method of controlling gall mites was assessed for the first time. The results in the present study showed that artificial defoliation was particularly effective in preventing the goji berry gall mite *A. pallida* infestation by facilitating leaf abscission to block nutrient supply. The method of controlling gall-forming pests also reduces the risk of product and environmental contamination by decreasing the use of pesticides.

## ACKNOWLEDGEMENTS

The authors acknowledge the goji berry grower Mr. Jun Mao and the undergraduate student Jun Yang for their kind help.

### Funding

This work was supported by the National Natural Science Foundation Project of China (No. 81673699 and 81470168) and the Chinese Academy of Medical Sciences Innovation Fund for Medical Science (No. 2016-12M-3-017). The funders had no role in study design, data collection and analysis, decision to publish, or preparation of the manuscript.

### Grant Disclosures

The following grant information was disclosed by the authors:
National Natural Science Foundation Project of China: 81673699, 81470168.
Chinese Academy of Medical Sciences Innovation Fund for Medical Science: 2016-12M-3-017.

### Competing Interests

The authors declare there are no competing interests.

### Author Contributions

- Jianling Li conceived and designed the experiments, performed the experiments, analyzed the data, prepared figures and/or tables, authored or reviewed drafts of the paper, approved the final draft.

- Sai Liu conceived and designed the experiments, performed the experiments, authored or reviewed drafts of the paper, approved the final draft.
- Kun Guo, Haili Qiao and Rong Xu performed the experiments, contributed reagents/materials/analysis tools, approved the final draft.
- Changqing Xu and Jun Chen conceived and designed the experiments, approved the final draft.

## Data Availability

The raw measurements are available in the Supplemental File.

## Supplemental Information

Supplemental information for this article can be found online at http://dx.doi.org/10.7717/peerj.6503#supplemental-information.

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
