# Peer review of "A new method of gall mite management: application of artificial defoliation to control Aceria pallida"

_PeerJ, doi:10.7717/peerj.6503_

## Round 0.1 · original submission · Major Revisions

Dear Dr. Li and colleagues:

Thanks for submitting your work to PeerJ. I apologize for the lengthy review process, as I have been waiting on one final review that is well overdue. I have decided to move forward with the three reviews that we have. Two of these reviewers have raised some concerns about the research. Despite this, these reviewers are optimistic about your work and the potential impact it will lend to research on gall mite management. Thus, I encourage you to revise your manuscript accordingly, taking into account all of the concerns raised by both reviewers.

Please note that reviewer 3 kindly provided a marked-up version of your manuscript.

Please also heed the suggestions for improvements to methods, tables and figures. It would also be a good idea to have an English expert proof your manuscript prior to resubmission.

I look forward to seeing your revision, and thanks again for submitting your work to PeerJ.

Good luck with your revision,

-joe

Reviewer 1 ·

Basic reporting

This paper gives new information to the eriophyid mites control.

Experimental design

is okay

Validity of the findings

are original;

Reviewer 2 ·

Basic reporting

This manuscript reports on a new and unique method of managing eriophyid mites; namely, defoliating the mite galls between natural fruit production/harvest periods to prevent the re-infestation of new plant growth. The structure of the manuscript is clear and unambiguous; methods, tables and figures need improvement. The Western world is familiar with wolfberry as goji berries; perhaps this could be included in the Introduction. Additionally from the appearance of the plots in Figure 3, you should refer to the plants as bushes.

Specific problems with the Introduction:
1. Line 42 By definition mites do not have wings; delete this.
2. L 48 LI et al. should be Li et al.
3. Throughout the manuscript delete the 's' from damages; i.e. damage caused by gall mites.
4. L 52 should be 'Furthermore'.
5. L 55 The citation of Greenberger et al., 2004 is not appropriate and doesn't support your hypothesis. Boll weevils are not attached or encased in cotton leaves; therefore, defoliating cotton leaves per se will not be effective management. In the cited manuscript, beetles were affected through direct contact spraying and in some combinations there was a synergistic effect between defoliant(s) and insecticide(s). In your methods, to be parallel to the Greenberger publication, you should have tested the direct effect of the defoliate on the mites. In contrast, Bemisia tabaci nymphs are attached to the leaves, so defoliating the leaves also defoliates the nymphs, which subsequently died due to lack of food.

Experimental design

The area/dimensions/number of bushes of each experimental treatment was not clearly stated. The area of the experimental plot (L 97) was given as 2600 sq m. There were five replicates of each pesticide treatment and defoliant with about 60 bushes as a buffer between treatment areas. The number of treated bushes is not given (L101-111).

L 112 states that two bushes per replicate and four branches were selected for gall counting. Two bushes out of how many? How many leaves per branch?

There is insufficient information, and no raw data to ascertain what was done.

Validity of the findings

1. The validity of the findings are impossible to know as the number of treated plants was not given. Additionally the degrees of freedom were also not given, therefore the number of treated plants could not be ascertained, even indirectly.
2. On both figures there are only two lines - 'defoliant' which was the same over the 3 years of the trials, and 'pesticides' where were different combinations for each spray date over 3 years; and counts were made twice a month. We should see these results for each treatment.
3. Figure 2 axes are: Number of ... per branch. However there were four branches per plant for each of two plants, so I expect to see +/- S.D. which is not shown on the graphs. Additionally was there a difference among the cardinal directions?
4. The 'raw data' files are not present, just the pooled data for making the graphs, so it is impossible to know what the data were.

Additional comments

The idea of defoliating is new and novel, and probably worked just fine. However, the data are not presented well and so conclusions are???

Reviewer 3 ·

Basic reporting

This is an original research. The investigation is relevant due to the economic impact of gall mite on the production of wolfberry, a tree of a great importance in traditional Chinese medicine and food.
Professional English was used but several errors in syntaxes were noticed. Particularly, Experimental design and Results phrasing makes comprehension difficult as well as the interpretation of data. Please ask for support in English writing.
The discussion seems quite weak with few references.

Experimental design

Research question is well defined. However, some important questions regarding experimental design emerged during the reading. Methods are not well described to replicate. Please see the .pdf file with comments.

Validity of the findings

No comment

Additional comments

Please check carefully the comments on manuscript. Thanks.

Annotated reviews are not available for download in order to protect the identity of reviewers who chose to remain anonymous.

---

## Round 0.2 · accepted · Accept

Dear Dr. Li and colleagues:

Thanks for re-submitting your manuscript to PeerJ, and for addressing the concerns raised by the reviewers. I now believe that your manuscript is suitable for publication. Congratulations! There a few minor items to address, per reviewer 2. Please handle these while in production. I look forward to seeing this work in print, and I anticipate it being an important resource for the communities studying gall mite management. Thanks again for choosing PeerJ to publish such important work.

Good luck with your revision,

-joe

# Reviewer 2 ·

Basic reporting

This manuscript is much improved over the previous version. There are some minor English editorial changes which, I'm sure, the production staff will catch and correct.

Experimental design

no comment

Validity of the findings

no comment

Additional comments

Line 53, line298 - what is correct date 2013 or 2003?

Line 122 - Tukey's HSD test not Turkey

The first paragraph of the Discussion is largely a repeat of the Introduction, and should be significantly reduced. The Discussion is for the results not repeating the introduction.

I see that the authors used only one type of defoliant year after year. I think something should be added in the discussion about practices to avoid the development of resistance to the defoliants, similar to IRM.

Line 205 Wiley et al. not in references

Line 249 Rhodiola rosea should be in italics
L 264 Oikos in italics

Figure 1 adult A. pallida observed using a scanning electron microscope

Figure 2 The arrows are actually white with a black outline. Maybe make them solid black to see them more easily.

D, E, and F, I believe are average number of gall/day

Reviewer 3 ·

Basic reporting

Professional Englished used. Sufficient background provided.

Experimental design

Methods described with sufficient detail & information to replicate.

Validity of the findings

No comment

Additional comments

Authors made an effort to support the methodology/results as asked in the first revision.Authors demonstrated a qualified revision of the language and improved the manuscript.. I am totally satisfied with the corrections. As a consequence, I accept the publication in Peer J.